# Extramedullary T-lymphoblastic Crisis in a Myelodysplastic/Myeloproliferative Neoplasm with a t(12;22)/*MN1::ETV6* Translocation

Ana Carolina Freitas [1,*], Tiago Maia [2], Joana Desterro [1], Francesca Pierdomenico [1], Albertina Nunes [1], Isabelina Ferreira [3], José Cabeçadas [2] and Maria Gomes da Silva [1]

1   Department of Hematology, Portuguese Institute of Oncology Lisbon, 1099-023 Lisbon, Portugal
2   Department of Pathology, Portuguese Institute of Oncology Lisbon, 1099-023 Lisbon, Portugal
3   Department of Bone Marrow Transplantation, Portuguese Institute of Oncology Lisbon, 1099-023 Lisbon, Portugal
*   Correspondence: aclfreitas@ipolisboa.min-saude.pt

**Abstract:** Myelodysplastic/myeloproliferative neoplasms (MDS/MPN) are not a single disease, but rather a heterogenous group of entities which are increasingly subclassified according to recurrent genetic abnormalities. Chromosomal translocations involving meningioma 1 (*MN1*) and ETS variant 6 (*ETV6*) genes are extremely rare, but recurrent in myeloid neoplasms. We describe the case of a patient with a myelodysplastic/myeloproliferative neoplasm with neutrophilia, who developed an extramedullary T-lymphoblastic crisis with the t(12;22)(p13;q12) translocation as the only cytogenetic abnormality. This case shares several clinical and molecular features with myeloid/lymphoid neoplasms with eosinophilia. The treatment of this patient was challenging, as the disease proved to be highly refractory to chemotherapy, with allogenic stem cell transplantation as the only curative option. This clinical presentation has not been reported in association with these genetic alterations and supports the concept of a hematopoietic neoplasm originating in an early uncommitted precursor cell. Additionally, it stresses the importance of molecular characterization in the classification and prognostic stratification of these entities.

**Keywords:** myelodysplastic/myeloproliferative neoplasm; T-lymphoblasts; *MN1* gene; *ETV6* gene; t(12;22)(p13;q12) translocation; myeloid/lymphoid neoplasms with eosinophilia

## 1. Introduction

Myelodysplastic/myeloproliferative neoplasms (MDS/MPN) are characterized by the coexistence of myeloproliferative features, with ineffective hematopoiesis and comprise a diverse group of entities, from the clinical, morphologic, genetic and prognostic points of view [1]. This heterogeneity is reflected in the 2022 World Health Organization and International Consensus (WHO/ICC) classification of myeloid malignancies, with the creation of new subcategories of MDS/MPN, mostly according to molecular alterations [2,3].

The use of molecular pathology to predict clinical behavior, support a diagnosis or define specific nosologic entities among myeloid neoplasms has been expanding. Besides the prototypical example of *BCR::ABL* fusion as a sine qua non condition for the diagnosis of chronic myeloid leukemia, an increasing number of tyrosine kinase gene fusions—*PDGFRA*, *PDGFRB*, *FGFR1*, *JAK2*, *ABL1*, *FLT3*—are recognized to diagnose myeloid/lymphoid neoplasms with eosinophilia [2]. *SF3B1* mutation is required to define a specific subgroup of MDS/MPN with thrombocytosis [2] and *CSF3R* mutation, although not required, strongly supports a diagnosis of chronic neutrophilic leukemia. Moreover, specific genetic alterations are now used to stratify the behavior of myelodysplastic neoplasms (MDS) with less than 20% myeloblasts, but rapid progression to acute myeloid leukemia (AML), reclassifying them as genetically defined AML [4].

Cases harboring rare molecular abnormalities, sometimes reported to be associated with aggressive clinical course, are still not recognized as distinct entities nor as specific prognostic groups within a given entity, which may originate challenges for prognostication and clinical management. This is the case of myeloid neoplasms associated with the t(12;22)(p13;q12), which involves ETS variant 6 (*ETV6*) and meningioma 1 (*MN1*) genes.

MN1 is still not fully understood from a functional point of view, but it appears to have transcriptional activator functions [5]. Overexpression of MN1 has been mostly reported in acute myeloid leukemia and appears to associate with a dismal prognosis and poor response to conventional chemotherapy [6–9]. On the other hand, *ETV6*, located on chromosome 12p13, is frequently involved in hemopoietic malignancies [10], and *ETV6*-rearranged neoplasms are increasingly recognized as defining entities. *ETV6* gene rearrangements involving tyrosine kinase partner genes are now found in the category of myeloid/lymphoid neoplasms with eosinophilia in the new classifications. Even so, the translocation of *MN1* gene with *ETV6* is a very rare event and is predominantly described in myeloid neoplasms. Most of these cases correspond to acute myeloid leukemias, but the rearrangement is also described in MDS, neoplasms with overlapping features of MDS and myeloproliferative neoplasm (MPN) and, rarely (two cases reported), in association with myeloid/T mixed-phenotype acute leukemias [11–15].

## 2. Case Report

We report the case of a 40-year-old man with a recent SARS-CoV2 infection and an otherwise unremarkable history. He was referred to the hematology department for suspected acute leukemia. The patient had been observed in his local hospital for asthenia, significant weight loss and cervical enlarged lymph nodes over the past month, without fever or other symptoms, suggesting infection. Blood tests showed marked leukocytosis (182.000/mm$^3$), bicytopenia (Hb 7.6 g/dL and platelets 52.000/mm$^3$) and elevated lactate dehydrogenase (680 UI/L). The peripheral blood smear (Figure 1A) revealed a marked increase in neutrophils (55%) and their precursors (>10%), as well as eosinophilia (6%).

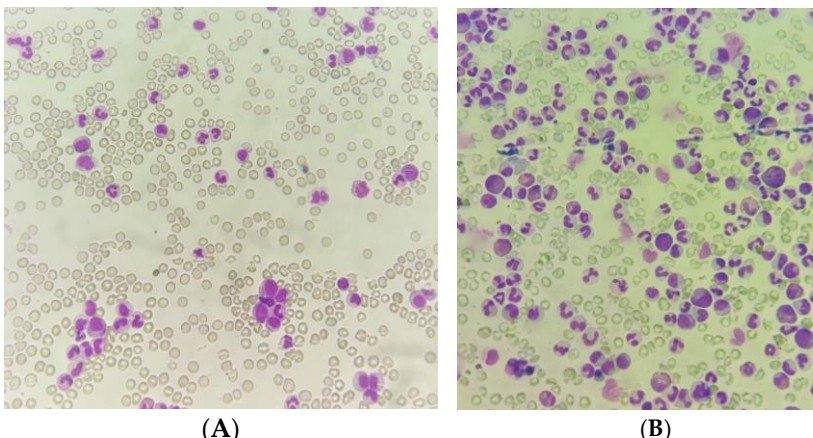

(**A**)  (**B**)

**Figure 1.** (**A**) Peripheral blood smear showing increased neutrophils and their precursors. (**B**) Bone marrow aspirate smear demonstrating myeloid hyperplasia and granulocytic dysplasia.

At admission, physical examination confirmed enlarged bilateral cervical and axillary lymph nodes, the largest one measuring at about 3 cm, and splenomegaly palpable 4 cm below the left costal margin. Bone marrow aspirate (Figure 1B) revealed a hypercellular bone marrow with myeloid hyperplasia, dysgranulopoiesis, and eosinophilia, without an increase in the number of blasts.

Conventional cytogenetic analysis (Figure 2) showed 12 metaphases with t(12;22)(p13;q12) and no additional chromosomal aberrations.

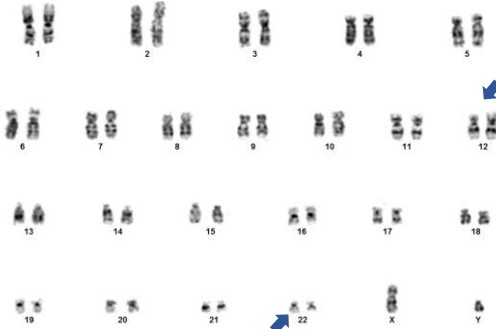

**Figure 2.** Patient's bone marrow karyotype, showing the t(12;22)(p13;q12) translocation (blue arrows).

The bone marrow trephine biopsy (Figure 3) was compatible with a myeloid neoplasm, with overlapping features of myelodysplastic and myeloproliferative neoplasm: a packed marrow, 100% cellular, with a myeloid-dominant cell component, with increased eosinophilic precursors, but preserved maturation. Additionally, dyserythropoiesis and dysmegakaryopoiesis were found, with no evidence of bone marrow fibrosis.

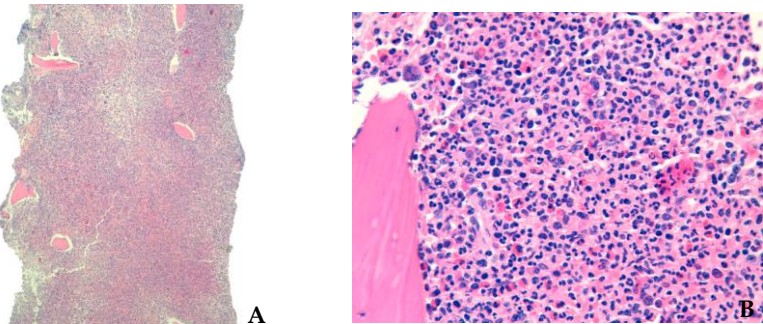

**Figure 3.** (**A**) Bone marrow trephine biopsy (low power), showing hypercellular bone marrow. (**B**) Bone marrow biopsy (medium power) stained with H&E, showing overlapping features of myelodysplastic and myeloproliferative neoplasia, with increased eosinophilic precursors.

Unexpectedly, the flow cytometry characterization of lymph node fine-needle aspiration revealed a population of lymphoid T cells expressing CD7, CD5, CD2, CD4, CD8, CD38 and CD45. This population was not identified in bone marrow flow cytometry. The cervical lymph node biopsy demonstrated extramedullary hematopoiesis similar to the morphology on the bone marrow trephine, together with an intermingled infiltration by T-cell lymphoblasts (Figure 4A,B). These features are better evidenced through immunohistochemical staining with myeloperoxidase, terminal deoxynucleotidyl transferase (TdT), CD99 and CD1a (Figure 4C,D). To confirm the clonality of the lymphoid population, the T-cell receptor rearrangements were studied in the lymph node aspirate and a clonal rearrangement for the beta genes was found.

In order to determine whether the cytogenetic alteration present in the bone marrow also occurred in the lymph node, we performed fluorescence in situ hybridization (FISH) studies, using a dual color break-apart probe for *ETV6* (12p13). Split signals, consistent with an *ETV6* rearrangement, were identified in large nuclei and in smaller nuclei that were presumed to correspond to myeloid cells and lymphoblasts, respectively (Figure 5A,B). This analysis was performed in an area of the lymph node rich in T lymphoblasts. The FISH probe showed similar split signals in the bone marrow aspirate (Figure 5C), supporting a possible clonal relationship.

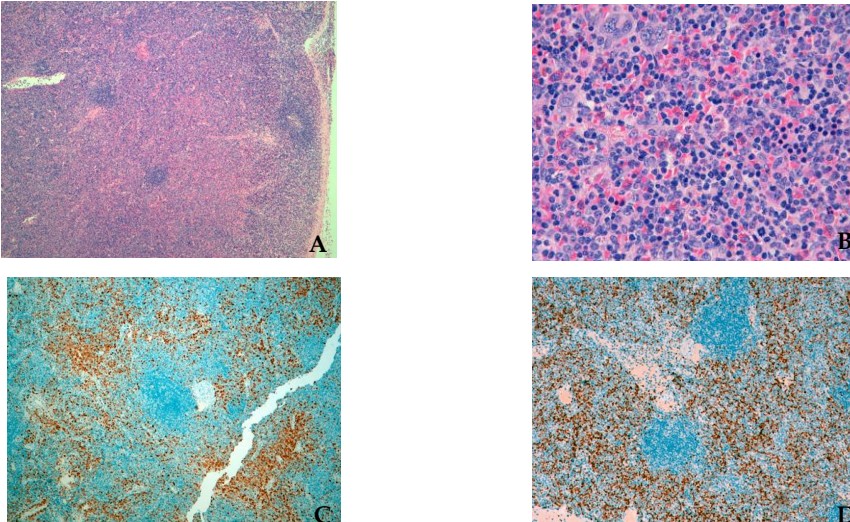

**Figure 4.** (**A**) Lymph node biopsy (medium power) stained with H&E with effaced architecture. (**B**) Lymph node biopsy (high power) showing extramedullary hematopoiesis with eosinophilia (similar to the morphology on the bone marrow trephine), and an intermingled proliferation of cells with blastoid morphology. (**C**) Immunohistochemical stain for myeloperoxidase shows a lymph node with extramedullary hematopoiesis. (**D**) Immunohistochemical stain of the lymph node biopsy for terminal deoxynucleotidyl transferase (TdT) shows an infiltration by T-cell lymphoblasts.

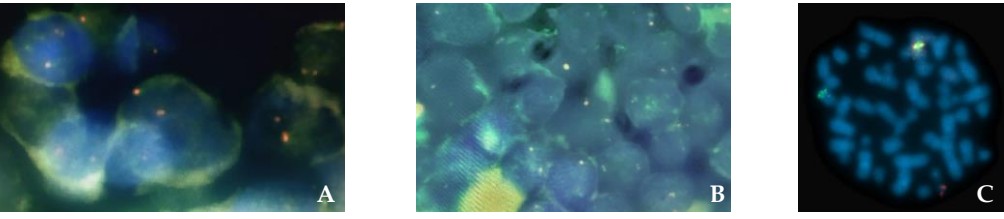

**Figure 5.** Fluorescence in situ hybridization study in the lymph node (**A**,**B**) and in the bone marrow, (**C**) using a dual color break-apart probe for *ETV6* (12p13). (**A**) Cells with larger nuclei, interpreted as myeloid cells, showing rearrangement of *ETV6* (12p13). (**B**) Cells with small/medium sized nuclei, interpreted as lymphoblasts, showing rearrangement of *ETV6* (12p13). (**C**) Fluorescence in situ hybridization study in the bone marrow, showing rearrangement of *ETV6*.

For further characterization from the biological point of view, we studied panel that included 54 genes (Table S1) related to myeloid neoplasms, including *ASXL1*, *BCOR*, *EZH2*, *SETBP1*, *SF3B1*, *SRSF2*, *STAG2*, *TET2* and *U2AF1* genes by next-generation sequencing (NGS). The genes associated with germline predisposition, such as *CEBPA*, *RUNX1*, *TP53* and *GATA2*, were also included in this analysis. No pathogenic variants were found, both in the bone marrow and in the lymph node aspirate.

In addition, defined and recurrent genetic abnormalities were excluded, such as BCR–ABL fusion, mutations in *JAK2V617F*, *MPL* and *CALR* and tyrosine kinase gene fusions associated with myeloid/lymphoid neoplasms with eosinophilia (*PDGFRA*, *PDGFRB* and *FGFR1*).

Taking all these data into account, the patient met criteria of MDS/MPN with neutrophilia (former atypical chronic myeloid leukemia), with an extramedullary T-lymphoblastic crisis.

Staging by 2-deoxy-2-[18F]fluoro-d-glucose Positron Emission Tomography (FDG-PET) showed involvement of cervical and axillary lymph nodes, with low intensity SUV, splenomegaly and diffuse intramedullary bone uptake.

Considering that the lymphoid component was clinically the most aggressive one, induction chemotherapy was started with the hyperfractionated cyclophosphamide, vincristine, doxorubicin and dexamethasone (hyper-CVAD) regimen [16]. This led to the

disappearance of the enlarged lymph nodes and also to a consistent improvement in the symptoms referred at presentation. However, progressive leukocytosis, splenomegaly, and persistence of the myelodysplastic/myeloproliferative component in the bone marrow were observed, along with de novo atypical mast cell proliferation (spindled, with CD25 immunohistochemical expression). This population of mast cells did not form aggregates and no mutations in the *KIT* gene were detected in the NGS analysis. We proceeded with a reinduction and consolidation with the FLAG-ida regimen (fludarabine, cytarabine, idarubicin, and G-CSF) [17,18], also without any effect on the myeloid compartment.

The patient underwent allogeneic hematopoietic stem cell transplantation from a fully matched, unrelated donor, 8 months after the diagnosis. He received myeloablative conditioning with high dose cyclophosphamide (120 mg/kg) and total body irradiation (12 Gy). On day +30, we observed resolution of myelodysplastic and myeloproliferative features in the bone marrow, but 2 months later, the mast cell proliferation was documented again, with a normal karyotype and complete donor chimerism. At this time, the serum tryptase levels were 20 ng/mL and the analysis of the *KIT* gene did not reveal mutations in exons 8 and 17. Eight months after the transplant, the cytogenetic alteration detected at diagnosis was identified, as well as a balanced translocation between the short arm of chromosome 3 (p24~25) and the long arm of chromosome 15 (q21~22). This anomaly was confirmed with FISH studies, with whole-chromosome painting probes for chromosomes 3 and 15: 46,XY, t(12;22)(p13;q12) [3]/46,idem,t(3;15)(p24~25;q21~22) [4]/46,XY [15]. ish t(3;15)(wcp15+,wcp13+;wcp15+,wcp3+) [2]. This additional cytogenetic alteration is suggestive of clonal evolution. At this time, a decrease in donor chimerism was also observed, leading to discontinuation of immunosuppression. Currently, 14 months after allogeneic hematopoietic stem cell transplantation, the patient is clinically stable, without any symptoms and with no evidence of disease progression in terms of blood counts. There was recovery of donor chimerism; however, the cytogenetic alteration continues to be detected, but without other signs of the underlying disease, namely, the mast cell population has not been demonstrated again. We decided to maintain a close surveillance, and if the disease progresses, therapy with 5-azacytidine plus venetoclax and subsequent donor lymphocyte infusion is planned.

## 3. Discussion

Hematopoietic neoplasms with t(12;22)(p13;q12) are rare, with about 40 published cases [11–15].These followed an aggressive clinical course, characterized by chemorefractoriness, regardless of the different phenotypes at presentation (AML, MDS, MDS/MPN). The cases in which progression to/presentation as acute leukemia was documented were mostly of myeloblastic type, with only two cases where presentation was as a myeloid/T mixed-phenotype acute leukemia. To the best of our knowledge, this is the first case describing an extramedullary T-lymphoblastic crisis in a myelodysplastic/myeloproliferative neoplasm, with this translocation as the only genetic alteration. This case supports the concept of a hematologic neoplasm probably originating in an early hematopoietic precursor, able to differentiate into both myeloid and lymphoid lineages. This fact may be associated with the genetic alteration inherent to this translocation, since early hematopoietic progenitors, but not granulocyte macrophage progenitors, seem to be prone to MN1-induced transformation in relation to their high levels of Hoxa9 and Meis1 expression [19]. In animal models, it has also been described that MN1::ETV6 expression can have non-lineage-specific leukemogenic effects, disturbing both the myeloid and lymphoid growth and causing immature T-cell neoplasms [20].

No recurrent genetic alterations were found, nor in other genes associated with myeloid neoplasms, including those related to MDS/MPN with neutrophilia. Considering the age of the patient and the clinical course, even with an unremarkable previous history, we could think of a germline predisposition, but this has not been demonstrated either. All these data favor the hypothesis that the only genetic abnormality found, t(12;22), is sufficiently important in the pathogenesis of this disease.

Interestingly, our case shares several features with myeloid/lymphoid neoplasms with eosinophilia: extramedullary proliferation, eosinophilia and atypical mast cell proliferation. Although *ETV6* was incorporated as a partner gene for tyrosine kinase receptors (*JAK2*, *PDGFRA*, *PDGFRB*, *FLT3*, *ABL1*) in the new WHO/ICC classifications [2,3], our case does not fulfill the strict molecular diagnostic criteria. In addition, in cases harboring t(12;22)(p13;q12), dysregulation of a poorly characterized tyrosine kinase receptor cannot be excluded, considering that some are neighbors of *ETV6* (*SYTK1*) or within the band region of *MN1*(*GRK3*). For classification and management purposes, our case overlaps with the concept of myeloid/lymphoid neoplasms with eosinophilia and, together with previous evidence, expands the genetic landscape of neoplasms exhibiting similar clinical and morpho-phenotypic features. Therefore, we raise the question of whether *ETV6* and *MN1* gene rearrangement should be tested in clinical and morphological presentations of hematologic neoplasms resembling myeloid/lymphoid neoplasms with eosinophilia that are negative for *JAK2*, *PDGFRA*, *PDGFRB*, *FLT3* and *ABL1* alterations.

Our patient was treated with the hyper-CVAD protocol, taking into account that the T-lymphoblastic lymphoma was the most aggressive clinical component. A lymph node response was observed, but the myelodysplastic/myeloproliferative neoplasia proved to be highly refractory to chemotherapy, either with hyper-CVAD or with FLAG-ida. In fact, the *MN1::ETV6* translocation seems to confer a poor prognosis and resistance to chemotherapy alone, which may be related to the expression of multidrug-resistance-associated genes in neoplastic hematopoietic cells [21]. Despite the dismal prognosis, patients bearing this rearrangement may benefit from allogeneic hematopoietic stem cell transplantation, as early as possible in the course of the disease. In our case, even after a transplant with a myeloablative conditioning, cytogenetic relapse was observed, proving unpredictable disease behavior.

Acute leukemia with this genetic alteration may share several clinical and, especially, molecular characteristics with leukemia with rearrangements involving the lysine methyltransferase 2A (*KMT2A*) gene. One of these features, which may be therapeutically important in the future, is the overexpression of Hoxa and Meis1 [22]. The expression of these genes may constitute a potential biomarker to predict the response to new drugs that are under development, such as menin inhibitors, as the menin binding site is a crucial co-factor for the activity of Hox gene promoters [23].

With this work, we describe, for the first time, the association of an extramedullary T-lymphoblastic crisis with a myelodysplastic/myeloproliferative neoplasm with t(12;22)(p13;q12). The unusual biological characteristics of the disease, its refractoriness to conventional chemotherapy and the ability to differentiate into two lineages is in line with what has been described in the literature for other hematological neoplasms with this translocation. Notwithstanding, t(12;22) is exceedingly rare in hematopoietic malignancies with a lymphoid component. Additional research will be valuable to better characterize the pathogenic mechanisms of this genetic alteration and identify possible therapeutic targets.

**Supplementary Materials:** The following supporting information can be downloaded at: https://www.mdpi.com/article/10.3390/hematolrep15010022/s1. Table S1: List of genes and exons included in the Next Generation Sequencing panel.

**Author Contributions:** A.C.F. and T.M. writing—original draft preparation; A.C.F., F.P., J.D., A.N. and I.F. care of patient; T.M., J.D. and J.C. diagnosis of patient; M.G.d.S. supervision. All authors contributed to writing, reviewing, and editing the article. All authors have read and agreed to the published version of the manuscript.

**Funding:** This research received no external funding.

**Institutional Review Board Statement:** Ethical review and approval were waived by the Institutional Review Board of the Portuguese Institute of Oncology of Lisbon due to the fact that it is a retrospective analysis of one case and was not included in a systematic investigation. There were no prospective interactions or interventions with the patient in order to publish the case report. Additionally, the patient provided informed consent, in accordance with the principles of the Declaration of Helsinki.

**Informed Consent Statement:** Written informed consent has been obtained from the patient to publish this paper.

**Data Availability Statement:** The data presented in this study are available on request from the corresponding author. The data are not publicly available due to privacy.

**Conflicts of Interest:** The authors declare no conflict of interest.

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
