# Peer review of "Extramedullary T-lymphoblastic Crisis in a Myelodysplastic/Myeloproliferative Neoplasm with a t(12;22)/MN1::ETV6 Translocation"

_hematolrep, doi:10.3390/hematolrep15010022_

Round 1
Reviewer 1 Report
GENERAL COMMENTS
Introduction is rather short and barely informative. It does not mention at all previously published similar cases. The case description is inadequate for the hematology-reader. As is usually the case, the authors need to show the initial peripheral blood and marrow aspirate smears of the described case in the routine May-Grunwald-Giemsa stain, demonstrating the described morphologic features, and then proceed to most deeply investigations. This is considered as a basic information for the reader/hematologist and such images are imperative. Moreover, previous literature has not been effectively reviewed. Even if published several decades ago, there are about 20 cases of MDS patients, who transformed to a lymphoblastic leukemia, published in the literature (see the link below), and some of them were of T-cell origin ALL. Moreover, a T-lymphoblastic blast crisis has been repeatedly described in patients with CML before the TKI era, indicating the potential of such an evolution in other stem cell diseases. Therefore, the presented case in this manuscript is not the first described in the literature, even when the previously reported cases have not been thoroughly investigated. These cases need to be reported, discussed and referenced in this manuscript. In general, there is a significant imbalance between patient data presented and discussion of the case in this manuscript. I suggest the authors, to enrich their case description with additional information, as those listed below and to attenuate their speculations discussed, since the majority, if not all of them are unproven speculations, not having been through an NGS study. To this point I think that even if the authors had not performed NGS analysis while their patient was alive, they should rather have performed it post-portem, in order to potentiate their nicely presented thoughts.
https://pubmed.ncbi.nlm.nih.gov/?term=myelodysplasctic+syndrome+terminating+as+lymphoblastic+leukemia&sort=date&size=200
SPECIFIC MAJOR ISSUES
Introduction
lines 7-10: “Other rare molecular abnormalities …” The authors should nominate here which molecular abnormalities they imply, or at least provide an example for some of them.
Case report
Lines 2-7: Please, confirm the presence or absence of fever at the initial presentation of the patients. Even when fever was absent this needs to be stated. Moreover please, provide more information for the severity of splenomegaly (cm below left costal margin and dimensions in U/S) and the dimensions of the enlarged lymph nodes.
Lines 11-12: Please, report how many metaphases were analyzed and in how many of them the mentioned abnormality has been observed.
Lines 13-17: Please, comment on the marrow percentage of RBC precursors and for the potential presence of BM fibrosis.
Lines 18-23: Please, report whether the expression of the classical myeloid antigens CD11b, CD13, CD14, CD15 and CD33 were tested or not, and how was their expression on the clonal cell population.
Lines 33-39: In view of the prominent mastocytosis the authors need to report whether serum tryptase levels have been measured or not and how it was.
Lines 42-44: Also here, the authors should make clear whether Allo-SCT has been performed in remission or with active disease. Moreover, they should report the dose of cyclophosphamide and of TBI in the conditioning regimen used.
Line 46: Since on day +90 a complete donor chimerism was detected what was the interpretation of the documented mastocytosis at that time point?.
Line 47-48: Please, clarify how did you define clonal evolution. Did you detect additional cytogenetic abnormalities? If yes, what kind of additional abnormalities have you identified?
Page 4, lines 1-2: Please, clarify what you mean by “there was a cytogenetic response”. In other words, describe completely the new cytogenetic findings.
Discussion
lines 6-8: “This case supports … into both, myeloid and lymphoid lineage.” Please, moderate this declaration, by adding “most probably” or “potentially” since you have not performed separate cytogenetic analysis from the peripheral blood lymphocytes and from marrow myeloid cells. The finding of common cytogenetic abnormality in the marrow and the lymph node is not a proof, because the lymph node might had been infiltrated by extramedullary myeloid components.
Lines 27-28: The authors need to clarify whether they have tested their patient for the presence of mutations of the genes they report here (JAK2, PDGFRα, PDGFRβ, FLT3 and ABL1).
Lines 29-35: Since the authors have correctly raised the probability of resistance to classical chemotherapeutic agents, they should comment on why they did not opt to add venetoclax in their regimen or in their preparatory conditioning regimen, before transplant, to render it more effective .
Figure 3:
Please, also provide a higher magnification lymph node image, preferably with one immunostain.
SPECIFIC MINOR ISSUES
Line 22: There is a verb missing in the secondary sentence starting with “or within the band…”
There are several missing commas throughout the manuscript. Please, incorporate.
Author Response
1- Introduction is rather short and barely informative. It does not mention at all previously published similar cases. The case description is inadequate for the hematology-reader. As is usually the case, the authors need to show the initial peripheral blood and marrow aspirate smears of the described case in the routine May-Grunwald-Giemsa stain, demonstrating the described morphologic features, and then proceed to most deeply investigations.
Thank you for raising this issue, we are in total agreement that we should have included an image of patient’s bone marrow aspirate. We changed the introduction and the case description according to the suggestions. Additionally, we added an image of the peripheral blood and bone marrow aspirate smear, as well as a more detailed description.
This information is presented in line 35 to 70 (changes in introduction), 79-80 (peripheral blood smear) and in figure 1A and 2A.
2- Moreover, previous literature has not been effectively reviewed. Even if published several decades ago, there are about 20 cases of MDS patients, who transformed to a lymphoblastic leukemia, published in the literature (see the link below), and some of them were of T-cell origin ALL. Moreover, a T-lymphoblastic blast crisis has been repeatedly described in patients with CML before the TKI era, indicating the potential of such an evolution in other stem cell diseases. Therefore, the presented case in this manuscript is not the first described in the literature, even when the previously reported cases have not been thoroughly investigated. These cases need to be reported, discussed and referenced in this manuscript.
We regret we haven’t been sufficiently clear. It was not our intention to say that this is the first case of lymphoblastic transformation of a myeloproliferative / myelodysplastic neoplasm, but rather that it is the first case with this presentation (myeloproliferative / myelodysplastic neoplasm with a T lymphoblastic crisis at initial presentation) with the t(12;22)(p13;q12) translocation. Without the translocation, we fully agree that the interest and rarity of the case becomes irrelevant. We believe we have not emphasized enough and indeed there was a sentence in the discussion that could be misleading. We have already made the necessary changes to the main text in accordance with the points raised.
3- I suggest the authors, to enrich their case description with additional information, as those listed below and to attenuate their speculations discussed, since the majority, if not all of them are unproven speculations, not having been through an NGS study. To this point I think that even if the authors had not performed NGS analysis while their patient was alive, they should rather have performed it post-portem, in order to potentiate their nicely presented thoughts.
We are grateful to the reviewer to bring the point about NGS study for discussion. First of all, we would like to clarify that the patient is alive and under surveillance as mentioned throughout the case description. Additionally, we strongly agree that we should have better described all the laboratory research we did, including the NGS. In fact, we performed at diagnosis an NGS panel that included 54 genes related to myeloid neoplasms (TruSight Myeloid Sequencing Panel®”), both in the medullary aspirate and in the lymph node aspirate. This panel was negative for all 54 genes analyzed. The panel included the following genes and exons: ABL1 (4-6), ASXL1 (12), ATRX (8-10, 17-31), BCOR, BCORL1, BRAF (15), CALR (9), CBL (8-9), CBLB (9-10), CBLC (9-10), CDKN2A, CEBPA, CSF3R (14-17), CUX1, DNMT3A, ETV6, EZH2, FBXW7 (9-11), FLT3 (14-16), GATA1 (2), GATA2 (2-6), GNAS (8-9), HRAS (2-3), IDH1 (4), IDH2 (4), IKZF1, JAK2 (12,14), JAK3 (13), KDM6A, KIT (2,8-11,13-17), KRAS (2,3), MLL (5-8), MPL (10), MYD88 (3-5), NOTCH1 (26-28,34), NPM1 (12), NRAS (2,3), PDGFRA (12,14,18), PHF6, PTEN (5,7), PTPN11 (3,13), RAD21, RUNX1, SETBP1, SF3B1 (13-16), SMC1A (2,11,16-17), SMC3 (10,13,19,23,25,28), SRSF2 (14), STAG2, TET2 (3-11), TP53 (2-11), U2AF1 (2,6), WT1 (7,9), ZRSR2.
Additionally, the rearrangements in the T cell receptor genes were studied by PCR and genescanning analysis, and a clonal rearrangement for the beta genes in the lymph node aspirate was found, supporting the lymphoid nature of the tumor population.
We also performed other molecular analyzes to exclude mutations or rearrangements of other genes, namely: BCR::ABL translocation, JAK2V617F gene mutation, mutations in exon 10 of the MPL gene, mutations in exon 9 of the CALR gene, PDGFRA gene rearrangement, PDGFRB gene rearrangement or FGFR1. These were all negative.
The only genetic alteration found is actually the one described throughout the case, and demonstrated by conventional cytogenetics and by FISH analysis.
This information is presented in line 208 to 209 and 262 to 269.
4- Introduction lines 7-10: “Other rare molecular abnormalities …” The authors should nominate here which molecular abnormalities they imply, or at least provide an example for some of them.
We fully agree with the comment and we have changed the text accordingly.
5-Case report Lines 2-7: Please, confirm the presence or absence of fever at the initial presentation of the patients. Even when fever was absent this needs to be stated. Moreover please, provide more information for the severity of splenomegaly (cm below left costal margin and dimensions in U/S) and the dimensions of the enlarged lymph nodes.
The patient did not have fever at diagnosis, the spleen was palpable 4 cm below the left costal margin and the larger lymph nodes had a diameter of 3 cm. We added this information in the case description (line 81-83)
6- Case report Lines 11-12: Please, report how many metaphases were analyzed and in how many of them the mentioned abnormality has been observed.
At diagnosis, in the medullary aspirate, 12 metaphases were studied and all showed a translocation between the short arm of chromosome 12 (p13) and the long arm of chromosome 22 (q12). These information was already in the text.
7-Case report Lines 13-17: Please, comment on the marrow percentage of RBC precursors and for the potential presence of BM fibrosis.
We have added this information in the main text in line with your comment.
8- Case report Lines 18-23: Please, report whether the expression of the classical myeloid antigens CD11b, CD13, CD14, CD15 and CD33 were tested or not, and how was their expression on the clonal cell population.
The analysis by flow cytometry of the bone marrow showed the myeloid lineage represented by 0.4% of immature cells (CD34 + / CD117 + / CD13 + / HLA-Dr +), in addition to 93.5% of cells with differentiation without phenotypic distortions according to CD11b evaluation /CD13 and CD117/HLA-Dr. The monocyte lineage (CD13++ / HLA-Dr+) was represented by 1.8% of cells with maturation.
There was no evidence of an immature T-cell lymphoid population in the bone marrow by flow cytometry, it was only detected in the lymph node.
9-Case Report Lines 33-39: In view of the prominent mastocytosis the authors need to report whether serum tryptase levels have been measured or not and how it was.
The serum tryptase levels were measured after transplant (about 60 days later) and were 20.20 ng/mL. No mutations in the KIT gene were both in bone marrow and in the lymph node aspirate, at diagnosis and after the transplant. We added this information to the case description (line 339-341).
10-Case report Lines 42-44: Also here, the authors should make clear whether Allo-SCT has been performed in remission or with active disease. Moreover, they should report the dose of cyclophosphamide and of TBI in the conditioning regimen used.
The transplant was performed with active myeloid disease, due to its chemorefractoriness as explained throughout the case description. The conditioning regimen included cyclophosphamide 60mg/kg x 2 days and TBI at a total dose of 12 Gy (6 fractions of 2 Gy). This information is presented in line 288 and 289.
11- Case report Line 46: Since on day +90 a complete donor chimerism was detected what was the interpretation of the documented mastocytosis at that time point?
We interpreted the presence of mast cells as evidence of disease. The analysis of chimerism (performed in the bone marrow aspirate) could have been limited by bone marrow fibrosis. As such, the aspirate may not correspond to the cells that actually exist in the bone marrow and were better observed in the biopsy. In favor of this observation is the fact that a few months later, a relapse was effectively observed. Additionally, despite the atypical phenotype, there were no other features suggestive of mastocytosis, such as the formation of aggregates. The tryptase dosage and absence of mutation in KIT are also in favor of this observation.
12- Case report Line 47-48: Please, clarify how did you define clonal evolution. Did you detect additional cytogenetic abnormalities? If yes, what kind of additional abnormalities have you identified?
At that time (8 months after transplantation), we repeated the bone marrow aspirate and 22 metaphases were studied, of which 15 did not present chromosomal abnormalities. The remaining 7 had t(12;22)(p13;q12); four of these metaphases additionally showed a balanced translocation between the short arm of chromosome 3 (p24~25) and the long arm of chromosome 15 (q21~22), this anomaly was confirmed with FISH studies with "Whole Chromosome Paint" probes for the chromosomes 3 and 15. The detection of new anomalies is suggestive of clonal evolution. We added this information to the clinical report (line 342 to 347).
13- Case report Page 4, lines 1-2: Please, clarify what you mean by “there was a cytogenetic response”. In other words, describe completely the new cytogenetic findings.
We reformulated this sentence according to your suggestions (line 351).
14- Discussion lines 6-8: “This case supports … into both, myeloid and lymphoid lineage.” Please, moderate this declaration, by adding “most probably” or “potentially” since you have not performed separate cytogenetic analysis from the peripheral blood lymphocytes and from marrow myeloid cells. The finding of common cytogenetic abnormality in the marrow and the lymph node is not a proof, because the lymph node might had been infiltrated by extramedullary myeloid components.
We modified the discussion according to the suggestions.
15- Lines 27-28: The authors need to clarify whether they have tested their patient for the presence of mutations of the genes they report here (JAK2, PDGFRα, PDGFRβ, FLT3 and ABL1).
This very pertinent question has already been answered in point 3 and the text has been amended by adding this important information (line 266 to 268).
16- Lines 29-35: Since the authors have correctly raised the probability of resistance to classical chemotherapeutic agents, they should comment on why they did not opt to add venetoclax in their regimen or in their preparatory conditioning regimen, before transplant, to render it more effective.
Before transplantation, there was no active acute leukemia component, only a myeloproliferative / myelodysplastic neoplasm, in an asymptomatic patient. Venetoclax was not approved for this indication in our country, at that time. As such it was not included in the treatment regimen.
17- Figure 3: Please, also provide a higher magnification lymph node image, preferably with one immunostain.
We added that image as requested (figure 4B).
18- SPECIFIC MINOR ISSUES Line 22: There is a verb missing in the secondary sentence starting with “or within the band…” There are several missing commas throughout the manuscript. Please, incorporate.
The text has been changed accordingly.
Reviewer 2 Report
The authors report an adult case of MDS/MPN with t(12;22)(p13;q12) with proliferation of lymphoid-lineage tumor cells possessing the same translocation in the lymph nodes. From this viewpoint, they suggest that the tumor may have emerged from a common myeloid/lymphoid precursor. This case deserves to be reported, because, as far as the reviewer know, such a case has not been reported in the past, and it is an interesting case in terms of considering the origin of the tumor cells.
The reviewer ask the authors to update the report according to my comments below.
Comment 1
It would be better to show data that suggest that tumor cells in lymph nodes are not consisted of single population that have markers of both granulocytic and T lymphocytic lineages but can be divided into two different types of cells. For example, a cell surface marker test could evidently indicate the presence of cells of the two lineages.
Comment 2
In addition to the bone marrow images of HE-stained specimen, a Giemsa-stained images of the bone marrow smear samples should be presented so that the abnormalities in cytomorphology can be clearly identified. If possible, smears of lymph nodes should be presented as well.
Comment 3
In this case presentation, chromosome examination (and FISH) demonstrated the presence of t(12;22)(p13;q12), but were other genetic abnormalities not considered? For example, were there no mutations or rearrangements in JAK2, CALR, MPL, etc.?
Although it is not necessarily required to show data, it would be possible to discuss the mechanism of the appearance of the two types of tumor cells in more detail if gene mutation analysis (by means of whole genome, whole exome, or panel sequence) is performed on both lymphocyte and granulocyte fractions. Then the mechanism of the emergence of the two types of tumor cells may be discussed in more detail.
Author Response
1- It would be better to show data that suggest that tumor cells in lymph nodes are not consisted of single population that have markers of both granulocytic and T lymphocytic lineages but can be divided into two different types of cells. For example, a cell surface marker test could evidently indicate the presence of cells of the two lineages.
We agree with you and we also have this analysis planned, through double labeling and repeating the FISH with the ETV6 probe. However, the lymph was involved by the myeloid neoplasia, as well as by the lymphoblasts. The split signal (break apart) was verified in the round, cohesive and intermediate sized nuclei (which we believe correspond to the lymphoblasts). Additionally, the FISH analysis was carried out in areas of the lymph node rich in lymphoblasts (bearing in mind that it was in these cells that we intended to demonstrate the rearrangement, which we already knew existed in the myeloid cells of the bone marrow). We know that it is a matter of probability and does not definitively prove the clonality between the two types of cells, but these are data in favor.
2- In addition to the bone marrow images of HE-stained specimen, a Giemsa-stained images of the bone marrow smear samples should be presented so that the abnormalities in cytomorphology can be clearly identified. If possible, smears of lymph nodes should be presented as well.
We appreciate the fact that you have raised this issue and we could not agree more. We have added an image of the peripheral blood and bone marrow smear that demonstrates these changes (figure 1A and 1B).
3-In this case presentation, chromosome examination (and FISH) demonstrated the presence of t(12;22)(p13;q12), but were other genetic abnormalities not considered? For example, were there no mutations or rearrangements in JAK2, CALR, MPL, etc.?
Although it is not necessarily required to show data, it would be possible to discuss the mechanism of the appearance of the two types of tumor cells in more detail if gene mutation analysis (by means of whole genome, whole exome, or panel sequence) is performed on both lymphocyte and granulocyte fractions. Then the mechanism of the emergence of the two types of tumor cells may be discussed in more detail.
We are grateful to the reviewer to bring this point for discussion.
In fact, we performed an NGS panel that included 54 genes related to myeloid neoplasms, both in the medullary aspirate and in the lymph node aspirate, at diagnosis. This panel was negative for all 54 genes analyzed. The panel is described in the point 3 in response to reviewer 1.
We also performed other molecular analyzes to exclude mutations or rearrangements of other genes, namely: BCR-ABL translocation, JAK2V617F gene mutation, mutations in exon 10 of the MPL gene, mutations in exon 9 of the CALR gene, PDGFRA gene rearrangement, PDGFRB gene rearrangement or FGFR1. These were all negative.
This information is presented in line 262 to 269.
Reviewer 3 Report
The authors present an interesting case in which it is difficult to establish a reliable diagnosis. Therefore, the need for integrated diagnosis in hematology is exemplified. In addition, the finding of t(12,22) involving ETV6 and MN1 is very interesting and infrequent. The finding of a limphoid neoplasm (acute lymphoblastic leukemia T in thymic cortical stage (CD1a+)) associated with a myeloid neoplasm (MDS/MPN) and in turn with a marked eosinophilia make one think, at least clinically, of a myeloid/lymphoid neoplasm with eosinophilia and ETV6 rearrangement, despite the fact that in the future WHO 2022 there is no mention of MN1::ETV6 rearrangement. This is because they only specifically mention ETV6 rearrangements with tyrosine kinase receptors and MN1 is a transcription factor.
Despite the interest of the case, I believe that it presents a series of deficits that I would like to highlight:
1) The diagnosis of MDS/MPN does not correspond to any specific diagnosis. In the new WHO classification, MDS/MPN are differentiated into: Chronic myelomonocytic leukaemia, Myelodysplastic/myeloproliferative neoplasm with neutrophilia (previously Atyical CML(aCML)), Myelodysplastic/myeloproliferative neoplasm with SF3B1 mutation and thrombocytosis, and Myelodysplastic/myeloproliferative neoplasm-NOS. The diagnostic workflow followed to support or rule out any of these entities is not presented in the article. In addition, only bone marrow biopsy is mentioned, but bone marrow aspirate is fundamental to try to establish a diagnosis. For example, the assessment of dysgranulopoiesis, dyserythropoiesis and dysmegakaryopoiesis should be done in the bone marrow aspirate. This information has not been provided by the authors. Granulocytic dysplasia, for example, is part of the diagnostic criteria for some of these neoplasms such as aCML. To reach the diagnosis of these entities, the analysis of peripheral blood morphology is also mandatory (e.g. to establish the presence of myelemia in aCML, which is a diagnostic criterion). Also no information has been given from the bone marrow flow cytometry study (again crucial the aspirate). Have you been able to rule out minimal bone marrow infiltration by T-lymphoblastic leukemia?
Could the patient meet the diagnostic criteria for aCMLL? Could the patient meet the diagnostic criteria for chronic eosinophilic leukemia? These entities and others should be adequately discussed.
2) The finding of CD25 expression on mast cells. While this is a finding described in myeloid/lymphoid neoplasms with eosinophilia, especially in PDGFRA and PDGFRB rearrangements, it is essential to rule out the presence of systemic mastocytosis with associated hematologic neoplasm. Serum tryptase? Does it present the D816V KIT mutation?
3) In view of the above points and given the diagnostic difficulties posed by the case, it seems to me mandatory to present information from an NGS panel in which the main mutations associated with myeloid neoplasms (ASXL1, SETBP1, TET2, SRSF2, ETNK1, ...) are studied. In addition, it would be very nice to observe the clonal relationship of myeloid and lymphoid neoplasia at the mutational level.
If these suggestions are not carried out, in my view the case is of more interest from a clinical than a diagnostic/biological point of view. In my opinion the case should be worked further from this point of view.
Thank you for the effort. I hope you find the revisions useful and that you will try to address as many of the points I suggest as possible.
Author Response
1- The diagnosis of MDS/MPN does not correspond to any specific diagnosis. In the new WHO classification, MDS/MPN are differentiated into: Chronic myelomonocytic leukaemia, Myelodysplastic/myeloproliferative neoplasm with neutrophilia (previously Atyical CML(aCML)), Myelodysplastic/myeloproliferative neoplasm with SF3B1 mutation and thrombocytosis, and Myelodysplastic/myeloproliferative neoplasm-NOS. The diagnostic workflow followed to support or rule out any of these entities is not presented in the article. In addition, only bone marrow biopsy is mentioned, but bone marrow aspirate is fundamental to try to establish a diagnosis. For example, the assessment of dysgranulopoiesis, dyserythropoiesis and dysmegakaryopoiesis should be done in the bone marrow aspirate. This information has not been provided by the authors. Granulocytic dysplasia, for example, is part of the diagnostic criteria for some of these neoplasms such as aCML. To reach the diagnosis of these entities, the analysis of peripheral blood morphology is also mandatory (e.g. to establish the presence of myelemia in aCML, which is a diagnostic criterion). Also no information has been given from the bone marrow flow cytometry study (again crucial the aspirate). Have you been able to rule out minimal bone marrow infiltration by T-lymphoblastic leukemia?
Could the patient meet the diagnostic criteria for aCMLL? Could the patient meet the diagnostic criteria for chronic eosinophilic leukemia? These entities and others should be adequately discussed.
We appreciate your comments and suggestions, they are very relevant. We have redrafted the text accordingly. Indeed, there are criteria for MDS/MPN with neutrophilia, both in peripheral blood and bone marrow aspirate (we have added that information). The molecular data did not support however the criteria is only “supportive and desirable criteria”.
This information is presented in line 270 to 272.
2- The finding of CD25 expression on mast cells. While this is a finding described in myeloid/lymphoid neoplasms with eosinophilia, especially in PDGFRA and PDGFRB rearrangements, it is essential to rule out the presence of systemic mastocytosis with associated hematologic neoplasm. Serum tryptase? Does it present the D816V KIT mutation?
We strongly agree that we should have specified these additional analyses better. The serum tryptase levels were measured after the transplant (about 60 days later) and were 20.20 ng/mL. No mutations in the KIT gene were detected in the NGS analysis, including the D816V. We added this information to the case description (line 339 to 341).
3- In view of the above points and given the diagnostic difficulties posed by the case, it seems to me mandatory to present information from an NGS panel in which the main mutations associated with myeloid neoplasms (ASXL1, SETBP1, TET2, SRSF2, ETNK1, ...) are studied. In addition, it would be very nice to observe the clonal relationship of myeloid and lymphoid neoplasia at the mutational level.
We are grateful to the reviewer to bring the point about NGS study for discussion.
We performed an NGS panel that included 54 genes related to myeloid neoplasms, both in the medullary aspirate and in the lymph node aspirate, at diagnosis. This panel was negative for all 54 genes analyzed. The panel is described in the point 3 in response to reviewer 1. This information has been added to the main text (line 262 to 269).
Regarding the clonal relationship, we totally agree with you and we also have this analysis planned, through double labeling and repeating the FISH with the ETV6 probe. However, the lymph was involved by the myeloid neoplasia, as well as by the lymphoblasts. The split signal (break apart) was verified in the round, cohesive and intermediate sized nuclei (which we believe correspond to the lymphoblasts). Additionally, the FISH analysis was carried out in areas of the lymph node rich in lymphoblasts (bearing in mind that it was in these cells that we intended to demonstrate the rearrangement, which we already knew existed in the myeloid cells of the bone marrow). We know that it is a matter of probability and does not definitively prove the clonality between the two types of cells, but these are data in favor.
Reviewer 4 Report
In this case report, the authors present a case who showed T-cell lymphoblastic crisis from MDS/MPN. The clinical course is interesting, and the manuscript is written well. Here are my comment that would improve the manuscript.
1. It is better to show the morphology of myeloid cells with atypia by bone marrow smears not by histology. I agree that the current histological findings clearly show the hyper cellular marrow, but it is hard to see the atypia of the cells.
2. Did the authors check the T-cell clone in the marrow?
3. Did the authors monitor the MN1::ETV6 after the stem cell transplantation?
Author Response
1- It is better to show the morphology of myeloid cells with atypia by bone marrow smears not by histology. I agree that the current histological findings clearly show the hyper cellular marrow, but it is hard to see the atypia of the cells.
We totally agree and appreciate you raising this point. We added an image of the peripheral blood and bone marrow smear (figure 1A and 1B).
2- Did the authors check the T-cell clone in the marrow?
We appreciate your question, and it is an important point. Upon diagnosis, we proceeded with an analysis by flow cytometry of the bone marrow aspirate, and the lymphocyte population did not show alterations suggestive of a clonal population, as opposed to what we found in the flow cytometry analysis of the lymph node.
Additionally, the T cell receptor rearrangements were studied in the lymph node aspirate and a clonal rearrangement for the beta genes was found.
We added this information to the case description (line 204).
- Did the authors monitor the MN1::ETV6 after the stem cell transplantation?
Cytogenetic monitoring was done and this data was added in more detailed. Since cytogenetic relapse was observed, we found no need to monitor by quantitative PCR.
Round 2
Reviewer 3 Report
The authors have addressed most of my requests adequately. However, I would appreciate one last effort from the authors:
1) Both the patient's age and the absence of mutations detected in their NGS panel could make us think of a neoplasm associated with the presence of germline mutations. Could they make a reasoned comment on this? In this regard, in the supplementary material, I would appreciate it if you could provide complete information on the panel used and, if possible, on the regions sequenced in the genes that comprise it. After that, it would be interesting to see which of the genes commonly related to myeloid neoplasms with germline mutations have not been analyzed to be able to comment on this as a possibility, given the extreme rarity of not finding mutations in a case like this.
Thank you very much for your effort, I think the case is very interesting and actually covers most of the aspects that needed to be covered.
Author Response
We are grateful to the reviewer to bring this point for discussion. Indeed, it is an important issue, and we have made changes to the text to address it more clearly. We studied all genes related to germline predisposition, with the exception of DDX41, ANKRD26 and SMAD9. But other important genes associated with germline predisposition to myeloid neoplasms, such as, CEBPA, RUNX1, TP53 and GATA2, were included in our analysis and no pathogenic variants were found both in the bone marrow and in the lymph node aspirate. We also emphasize that the patient had no alterations in the blood tests carried out years before the diagnosis (namely thrombocytopenia). Also, there were no phenotypic changes on physical examination, that would suggest a syndrome.
The fact that we did not find other genetic alterations known to be associated with myeloid neoplasms (germline or not), suggests that this translocation may be sufficiently important in the pathogenesis and progression of this disease.
As suggested, we have added a table (S1) in the supplementary material that details the genes and respective exons analyzed in the NGS panel.
We added this information in the main text (lines 325-327 and lines 440-443).